# SEMI-SUPERVISED LEARNING
# VIA NEW DEEP NETWORK INVERSION

## ABSTRACT

We exploit a recently derived inversion scheme for arbitrary deep neural networks to develop a new semi-supervised learning framework that applies to a wide range of systems and problems. The approach reaches current state-of-the-art methods on MNIST and provides reasonable performances on SVHN and CIFAR10. Through the introduced method, residual networks are for the first time applied to semi-supervised tasks. Experiments with one-dimensional signals highlight the generality of the method. Importantly, our approach is simple, efficient, and requires no change in the deep network architecture.

## 1 INTRODUCTION

Deep neural networks (DNNs) have made great strides recently in a wide range of difficult machine perception tasks. They consist of parametric functionals $f_\Theta$ with internal parameters $\Theta$. However, those systems are still trained in a *fully supervised* fashion using a large set of labeled data, which is tedious and costly to acquire.

*Semi-supervised learning* relaxes this requirement by leaning $\Theta$ based on two datasets: a labeled set $\mathcal{D}$ of $N$ training data pairs and an unlabeled set $\mathcal{D}_u$ of $N_u$ training inputs. Unlabeled training data is useful for learning as unlabeled inputs provide information on the statistical distribution of the data that can both guide the learning required to classify the supervised dataset and characterize the unlabeled samples in $\mathcal{D}_u$ hence improve generalization. Limited progress has been made on semi-supervised learning algorithms for DNNs Rasmus et al. (2015); Salimans et al. (2016); Patel et al. (2015), but today's methods suffer from a range of drawbacks, including training instability, lack of topology generalization, and computational complexity.

In this paper, we take two steps forward in semi-supervised learning for DNNs. First, we introduce an universal methodology to equip any deep neural net with an inverse that enables input reconstruction. Second, we introduce a new semi-supervised learning approach whose loss function features an additional term based on this aforementioned inverse guiding weight updates such that information contained in unlabeled data are incorporated into the learning process. Our key insight is that the defined and general inverse function can be easily derived and computed; thus for unlabeled data points we can both compute and minimize the error between the input signal and the estimate provided by applying the inverse function to the network output without extra cost or change in the used model. The simplicity of this approach, coupled with its universal applicability promise to significantly advance the purview of semi-supervised and unsupervised learning.

### 1.1 RELATED WORK

The standard approach to DNN inversion was proposed in Dua & Gupta (2000), and the only DNN model with reconstruction capabilities is based on autoencoders Ng (2011). While more complex topologies have been used, such as stacked convolutional autoencoder Masci et al. (2011), there

exists two main drawbacks. Firstly, the difficulty to train complex models in a stable manner even when using per layer optimization. Secondly, the difficulty to leverage supervised information into the learning of the representation.

The problem of semi-supervised learning with DNN has been attempted by several groups. Firstly, GAN based methods are a natural way to tackle this problem. The improved *generative adversarial network* (GAN) technique Salimans et al. (2016) couples two deep networks: a generative model that can create new signal samples, and a discriminative network. Both neural nets are trained jointly as in typical GAN framework. *Triple Generative Adversarial Nets* Li et al. (2017) propose a extension of the GAN framework for the particular task of semi-supervised by introduction of a third player. The task thus becomes simpler as there exist one discriminator labeling images (fake or real) and another predicting if the couples (image,label) are fake or not. Through this, better stability is reached. Finally, *Good Semi-supervised Learning That Requires a Bad GAN* Dai et al. (2017) currently hold SOTA method. This work lessen the same problem of GAN for semi-supervised of Li et al. (2017) by deriving analytical conditions and better formulation for the GAN objective hence providing a finer loss function as opposed to a third network.

The *semi-supervised with ladder network* approach Rasmus et al. (2015) employs a per-layer denoising reconstruction loss, which enables the system to be viewed as a stacked denoising autoencoder which is a standard and until now only way to tackle unsupervised tasks. By forcing the last denoising autoencoder to output an encoding describing the class distribution of the input, this deep unsupervised model is turned into a semi-supervised model. The main drawback of this method is the lack of a clear path to generalize it to other network topologies, such as recurrent or residual networks. Also, the per-layer "greedy" reconstruction loss might be too restrictive unless correctly weighted pushing the need for a precise and large cross-validation of hyper-parameters. The probabilistic formulation of deep convolutional nets presented in Patel et al. (2015) natively supports semi-supervised learning. The main drawbacks of this approach lies in the requirement that the activation functions be ReLU and that the overall network topology follows a deep convolutional network. Temporal Ensembling for Semi-Supervised Learning Laine & Aila (2016) propose to constrain the representations of a same input stimuli to be identical in the latent space despite the presence of dropout noise. This search of stability in the representation is analogous to the one of a siamese network Hoffer & Ailon (2015) but instead of presenting two different inputs, the same is used through two different models (induced by dropout). This technique provides an explicit loss for the unsupervised examples leading to the $\Pi$ model just described and a more efficient method denoted as temporal ensembling. Distributional Smoothing with Virtual Adversarial Training Miyato et al. (2015) proposes also a regularization term contraining the regularity of the DNN mapping for a given sample. Based on this a semi-supervised setting is derived by imposing for the unlabeled samples to maintain a stable DNN. Those two last described methods are the closest one of the proposed approach in this paper for which, the DNN stability will be replaced by a reconstruction ability, closely related to the DNN stability.

## 1.2 Contributions

This paper makes two main contributions: First, *we propose a simple way to invert any piecewise differentiable mapping, including DNNs.* We provide a formula for the inverse mapping of any DNN that requires no change to its structure. The mapping is computationally optimal, since the input reconstruction is computed via a backward pass through the network (as is used today for weight updates via backpropagation). Second, *we develop a new optimization framework for semi-supervised learning that features penalty terms that leverage the input reconstruction formula.* A range of experiments validate that our method improves significantly on the state-of-the-art for a number of different DNN topologies.

## 2 DEEP NEURAL NETWORK INVERSION

In this section we review the work of Balestriero & Baraniuk (2017) aiming at interpreting DNNs as linear splines. This interpretation provides a rigorous mathematical justification for the reconstruction in the context of deep learning.

### 2.1 DEEP NEURAL NETS AS LINEAR SPLINE OPERATORS

Recent work in Balestriero & Baraniuk (2017) demonstrated that DNNs of many topologies are or can be approximated arbitrary closely by multivariate linear splines. The upshot of this theory for this paper is that it enables one to easily derive an explicit input-output mapping formula. As a result, DNNs can be rewritten as a linear spline of the form

$$f_\Theta(x) = A[x]X_n + b[x], \tag{1}$$

where we denote by $f_\Theta$ a general DNN, $x$ a generic input, $A[x], b[x]$ the spline parameters conditioned on the input induced activations. Based on this interpretation, DNNs can be considered as template matching machines, where $A[x]$ plays the role of an input-adaptive template.

To illustrate this point we provide for two common topologies the exact input-output mappings. For a standard deep convolutional neural network (DCN) with succession of convolutions, nonlinearities, and pooling, one has

$$\boldsymbol{z}_{CNN}^{(L)}(x) = \underbrace{W^{(L)} \left[ \prod_{\ell=L-1}^{1} A_\rho^{(\ell)} A_\sigma^{(\ell)} \boldsymbol{C}^{(\ell)} \right] x}_{\text{Template Matching}} +$$

$$\underbrace{W^{(L)} \sum_{\ell=1}^{L-1} \left( \prod_{j=L-1}^{\ell+1} A_\rho^{(j)} A_\sigma^{(j)} \boldsymbol{C}^{(j)} \right) \left( A_\rho^{(\ell)} A_\sigma^{(\ell)} b^{(\ell)} \right) + b^{(L)}}_{\text{Bias}}, \tag{2}$$

where $\boldsymbol{z}^{(\ell)}(x)$ represents the latent representation at layer $\ell$ for input $x$. The total number of layers in a DNN is denoted as $L$ and the output of the last layer $\boldsymbol{z}^{(L)}(x)$ is the one before application of the softmax application. After application of the latter, the output is denoted by $\hat{y}(x)$. The product terms are from last to first layer as the composition of linear mappings is such that layer 1 is applied on the input, layer 2 on the output of the previous one and so on. The bias term results simply from the accumulation of all of the per-layer biases after distribution of the following layers' templates. For a Resnet DNN, one has

$$\boldsymbol{z}_{RES}^{(L)}(x) = \underbrace{W^{(L)} \left[ \prod_{\ell=L-1}^{1} \left( A_{\sigma,in}^{(\ell)} \boldsymbol{C}_{in}^{(\ell)} + \boldsymbol{C}_{out}^{(\ell)} \right) \right] x}_{\text{Template Matching}} +$$

$$\underbrace{\sum_{\ell=1}^{L-1} \left( \prod_{i=L-1}^{\ell+1} (A_{\sigma,in}^{(\ell)} \boldsymbol{C}_{in}^{(\ell)} + \boldsymbol{C}_{out}^{(\ell)}) \right) \left( A_{\sigma,in}^{(\ell)} b_{in}^{(\ell)} + b_{out}^{(\ell)} \right) + b^{(L)}}_{\text{Bias}}. \tag{3}$$

We briefly observe the differences between the templates of the two topologies. The presence of an extra term in $\prod_{\ell=L-1}^{1} \left( A_{\sigma,in}^{(\ell)} \boldsymbol{C}_{in}^{(\ell)} + \boldsymbol{C}_{out}^{(\ell)} \right)$ as opposed to $\prod_{\ell=L-1}^{1} A_\rho^{(\ell)} A_\sigma^{(\ell)} \boldsymbol{C}^{(\ell)}$ provides stability and a direct linear connection between the input $x$ and all

of the inner representations $z^{(\ell)}(x)$, hence providing much less information loss sensitivity to the nonlinear activations.

Based on those findings, and by imposing a simple $\ell_2$ norm upper bound on the templates, it has been shown that the optimal templates DNNs to perform prediction has templates proportional to the input, positively for the belonging class and negatively for the others Balestriero & Baraniuk (2017). This way, the loss cross-entropy function is minimized when using softmax final nonlinearity. Note that this result is specific to this setting. For example in the case of spherical softmax the optimal templates become null for the incorrect classes of the input.

**Theorem 1.** *In the case where all inputs have identity norm* $||X_n|| = 1, \forall n$ *and assuming all templates denoted by* $A[X_n]_c, c = 1, \ldots, C$ *have a norm constraint* $\sum_{c=1}^{C} ||A[X_n]_c||^2 \leq K, \forall X_n$, *then the unique globally optimal templates are*

$$
A^*[X_n]_c = \begin{cases} \sqrt{\frac{C-1}{C}K}X_n, \text{ if } c = Y_n \\ -\sqrt{\frac{K}{C(C-1)}}X_n, \text{ else.} \end{cases} \tag{4}
$$

We now leverage the analytical optimal DNN solution to demonstrate that reconstruction is indeed implied by such an optimum.

## 2.2    OPTIMAL DNN LEADS TO INPUT RECONSTRUCTION

Based on the previous analysis, it is possible to draw implications based on the theoretical optimal templates of DNNs. This is formulated through the corollary below. First, we propose the following inverse of a DNN as

$$
\begin{aligned}
f_\Theta^{-1}(X_n) &= A[X_n]^T (A[X_n]X_n + b[X_n]) \\
&= \sum_{c=1}^{C} (\langle A[X_n]_c, X_n \rangle + b[X_n]_c) A[X_n]_c.
\end{aligned} \tag{5}
$$

Following the analysis from a spline point of view, this reconstruction is leveraging the closest input hyperplane, found through the forward step, to represent the input. As a result this method provides a reconstruction based on the DNN representation of its input and should be part away from the task of exact input reconstruction which is an ill-posed problem in general.

The bias correction present has insightful meaning when compared to known frameworks and their inverse. In particular, when using ReLU based nonlinearities we will see that this scheme can be assimilated to a composition of soft-thresholding denoising technique. We present further details in the next section where we also provide ways to efficiently invert a network as well as describing the semi-supervised application.

## 2.3    IMPLEMENTATION AND NEW LOSS FUNCTION FOR SEMI-SUPERVISED LEARNING

We now apply the above inverse strategy to a given task with an arbitrary DNN. As exposed earlier, all the needed changes to support semi-supervised learning happen in the objective training function by adding extra terms. In our application, we used automatic differentiation (as in TheanoBergstra et al. (2010) and TensorFlowAbadi et al. (2016)). Then it is sufficient to change the objective loss function, and all the updates are adapted via the change in the gradients for each of the parameters. The efficiency of our inversion scheme is due to the fact that any deep network can be rewritten as a linear mapping Balestriero & Baraniuk (2017). This leads to a simple derivation of a network inverse

defined as $f^{-1}$ that will be used to derive our unsupervised and semi-supervised loss function via

$$
\begin{aligned}
f^{-1}(x, A[x], b[x]) &= A[x]^T \left( A[x]x + b[x] \right) \\
&= A[x]^T f(x; \Theta) \\
&= \frac{df(x; \Theta)}{dx}^T f(x; \Theta).
\end{aligned}
\tag{6}
$$

The main efficiency argument thus comes from

$$
A[x] = \frac{df(x; \Theta)}{dx},
\tag{7}
$$

which enables one to efficiently compute this matrix on any deep network via differentiation (as it would be done to back-propagate a gradient, for example). Interestingly for neural networks and many common frameworks such as wavelet thresholding, PCA, etc., the reconstruction error as $(\frac{df(x)}{dx})^T f(x)$ is the definition of the inverse transform. For illustration purposes, Tab. 1 gives some common frameworks for which the reconstruction error represents exactly the reconstruction loss.

Table 1: Examples of frameworks with similar inverse transform definition.

| | $\alpha_i$ | $f(x)_i$ | loss |
|---|---|---|---|
| Sparse Coding | Learned | $\frac{<x, W_i>}{||W_i||^2}$ | $||x - \sum_i \alpha_i \frac{df(x)_i}{dx}||^2 + \lambda||\alpha||_1$ |
| NMF | Learned | $< x, W_i >$ | $||x - \sum_i \alpha_i \frac{df(x)_i}{dx}||^2$ s.t. $J_f(x) \geq 0$ |
| PCA | $f(x)_i$ | $< x, W_i >$ | $||x - \sum_i \alpha_i \frac{df(x)_i}{dx}||^2$ s.t. $J_f(x)$ orth. |
| Soft Wavelet Thres. | $f(x)_i$ | $\max\left( | < x, W_i > | - b_i, 0 \right) sig(< x, W_i >)$ | $||x - \sum_i \alpha_i \frac{df(x)_i}{dx}||^2$ |
| Hard Wavelet Thres. | $f(x)_i$ | $1_{|<x, W_i>| - b_i > 0} < x, W_i >$ | $||x - \sum_i \alpha_i \frac{df(x)_i}{dx}||^2$ |
| Best Basis (WTA) | $f(x)_i$ | $1_{i = \mathrm{argmax}_i \frac{<x, W_i>}{||W_i||^2}} < x, W_i >$ | $||x - \sum_i \alpha_i \frac{df(x)_i}{dx}||^2$ |
| k-NN | 1 | $1_{i = \mathrm{argmax} <x, W_i> - ||W_i||^2 / 2} < x, W_i >$ | $||x - \sum_i \alpha_i \frac{df(x)_i}{dx}||^2$ |

We now describe how to incorporate this loss for semi-supervised and unsupervised learning. We first define the reconstruction loss $R$ as

$$
R(X_n) = \left\| \left( \frac{df_\Theta(X_n)}{dX_n} \right)^T f_\Theta(X_n) - X_n \right\|^2.
\tag{8}
$$

While we use the mean squared error, any other differentiable reconstruction loss can be used, such as cosine similarity. We also introduce an additional "specialization" loss defined as the Shannon entropy of the class belonging probability prediction

$$
E(\hat{y}(X_n)) = -\sum_{c=1}^{C} \hat{y}(X_n)_c \log(\hat{y}(X_n)_c).
\tag{9}
$$

This loss is intuitive and complementary to the reconstruction for the semi-supervised task. In fact, it will force a clustering of the unlabeled examples toward one of the clusters learned from the supervised loss and examples. We provide below experiments showing the benefits of this extra-term. As a result, we define our complete loss function as the combination of the standard cross entropy loss for labeled data denoted by $L_{CE}(Y_n, \hat{y}(X_n))$, the reconstruction loss, and the entropy loss as

$$
\mathcal{L}(X_n, Y_n) = 1_{\{Y_n \neq \emptyset\}} \alpha L_{CE}(Y_n, \hat{y}(X_n)) + (1 - \alpha)[\beta R(X_n) + (1 - \beta)E(X_n)1_{\{Y_n = \emptyset\}}],
\tag{10}
$$

with $\alpha, \beta \in [0, 1]^2$. The parameters $\alpha, \beta$ are introduced to form a convex combination of the losses, with $\alpha$ controlling the ratio between supervised and unsupervised loss and $\beta$ the ratio between the two unsupervised losses. This weighting is important, because the correct combination of the supervised and unsupervised losses will guide learning toward a better optimum (as we now demonstrated via experiments).

## 3 EXPERIMENTAL VALIDATION

### 3.1 SEMI-SUPERVISED EXPERIMENTS

We now present results of our approach on a semi-supervised task on the MNIST dataset, where we are able to obtain reasonable performances with different topologies. MNIST is made of 70000 grayscale images of shape $28 \times 28$ which is split into a training set of 60000 images and a test set of 10000 images. We present results for the case with $N_L = 50$ which represents the number of samples from the training set that are labeled and fixed for learning. All the others samples form the training set are unlabeled and thus used only with the reconstruction and entropy loss minimization. We perform a search over $(\alpha, \beta) \in \{0.3, 0.4, 0.5, 0.6, 0.7\} \times \{0.2, 0.3, 0.5\}$. In addition, 4 different topologies are tested and, for each, mean and max pooling are tested as well as inhibitor DNN (IDNN) as proposed in Balestriero & Baraniuk (2017). The latter proposes to stabilize training and remove biases units via introduction of winner-share-all connections. As would be expected based on the templates differences seen in the previous section, the Resnet topologies are able to reach the best performance. In particular, wide Resnet is able to outperform previous state-of-the-art results.

Running the proposed semi-supervised scheme on MNIST leads to the results presented in Tab. 2. We used the Theano and Lasagne libraries; and learning procedures and topologies are detailed in the appendix.The column 1000 corresponds to the accuracy after training of DNNs using only the supervised loss ($\alpha = 1, \beta = 0$) on 1000 labeled data. Thus, one can see the gap reached with the same network but with a change of loss and 20 times less labeled data.

We further present performances on CIFAR10 with 4000 labeled data in Tab. 3 and SVHN with 500 labeled data in Tab. 4. For both tasks we constrain ourselves to a deep CNN models, similar as the LargeCNN of Patel et al. (2015). Also, one of the cases correspond to the absence of entropy loss when $\beta = 1$. Furthermore to further present the generalization of the inverse technique we provide results with the leaky-ReLU nonlinearity as well as the sigmoid activation function.

MNIST Test Set Reconstruction After Training



Figure 1: Reconstruction of the studied DNN models for Resnet2-32 test set samples. The columns from left to right correspond to: the original image, mean-pooling reconstruction, max-pooling reconstruction, inhibitor connections.

### 3.2 SUPERVISED REGULARIZATION FOR IMPROVED GENERALIZATION

We now present and example of our approach on a supervised task on audio database (1D). It is the Bird10 dataset distributed online and described in Glotin et al. (2017). The task is to classify 10 bird species from their songs recorded in tropical forest. It is a subtask of the BirdLifeClef challenge.

Table 2: Accuracy on MNIST test set for 50 labeled examples in the training. The column $N_L = X$ gives the accuracy of the same networks trained with a $X$ labeled samples and the remaining of the training set as unlabeled examples.

| $N_L$ | 50, $(\alpha, \beta)$ | 100, | Sup1000, $(\alpha = 1, \beta = 0)$ |
|---|---|---|---|
| SmallCNNmean | 99.07,(0.7, 0.2) | - | 94.9 |
| SmallCNNmax | 98.63,(0.7, 0.2) | - | 95.0 |
| SmallUCNN | 98.85,(0.5, 0.2) | - | 96.0 |
| LargeCNNmean | 98.63,(0.6, 0.5) | - | 94.7 |
| LargeCNNmax | 98.79,(0.7, 0.5) | - | 94.8 |
| LargeUCNN | 98.23,(0.5, 0.5) | - | 96.1 |
| Resnet2-32mean | 99.11,(0.7, 0.2) | - | 95.5 |
| Resnet2-32max | **99.14**,(0.7, 0.2) | - | 94.9 |
| UResnet2-32 | 98.84,(0.7, 0.2) | - | 95.6 |
| Resnet3-16mean | 98.67,(0.7, 0.2) | - | 95.4 |
| Resnet3-16max | 98.56,(0.5, 0.5) | - | 94.8 |
| UResnet3-16 | 98.7,(0.7, 0.2) | - | 95.5 |
| (Mean,Std) | $(98.768, 0.249)$ | - | $(95.266, 0.462)$ |
| Improved GAN Salimans et al. (2016) | $97.79 \pm 1.36$ | $99.07 \pm 0.065$ | - |
| Auxiliary Deep Generative Model Maaløe et al. (2016) | - | 99.04 | - |
| LadderNetwork Rasmus et al. (2015) | - | $98.94 \pm 0.37$ | - |
| Skip Deep Generative Model Maaløe et al. (2016) | - | 98.68 | - |
| Virtual Adversarial Miyato et al. (2015) | - | 97.88 | - |
| catGAN Springenberg (2015) | - | $98.61 \pm 0.28$ | - |
| DGN Kingma et al. (2014) | - | $96.67 \pm 0.14$ | - |
| DRMM Patel et al. (2015) | — | 87.97 | - |
| Triple GAN Li et al. (2017) | - | $99.09 \pm 0.58$ | - |
| Semi-Sup Requires a Bad GAN Dai et al. (2017) | - | **99.2**$\pm$0.09 | - |

Table 3: Accuracy on CIFAR10 test set for $4000$ labeled examples in the training. Presented results include the average and standard deviation over 5 runs with random sampling of the labeled examples among the training set.

| $N_L$ | ReLU-CIFAR**4000**, | Sigmoid-CIFAR**4000**, |
|---|---|---|
| (0.7,0.1) | $76.97 \pm 0.39$ | $74.81 \pm 0.29$ |
| (0.7,0.2) | **77.75**$\pm$0.25 | **75.25**$\pm$0.31 |
| (0.7,0.3) | $76.42 \pm 0.43$ | $74.39 \pm 0.42$ |
| (0.8,0.1) | $76.77 \pm 0.32$ | $75.02 \pm 0.49$ |
| (0.8,0.2) | $76.62 \pm 0.33$ | $75.08 \pm 0.3$ |
| (0.8,0.3) | $76.98 \pm 0.07$ | $74.44 \pm 0.09$ |
| (0.85,1) | $75.28 \pm 0.2$ | $73.05 \pm 0.4$ |

| $N_L$ | CIFAR**4000**, | CIFAR**8000**, |
|---|---|---|
| Improved GAN Salimans et al. (2016) | $81.37 \pm 2.32$ | $82.28 \pm 1.82$ |
| LadderNetwork Rasmus et al. (2015) | $79.6 \pm 0.47$ | - |
| catGAN Springenberg (2015) | $80.42 \pm 0.46$ | - |
| DRMM +KL penalty Patel et al. (2015) | 76.76 | - |
| Triple GAN Li et al. (2017) | $83.01 \pm 0.36$ | - |
| Semi-Sup Requires a Bad GAN Dai et al. (2017) | $85.59 \pm 0.30$ | - |
| ΠModel Laine & Aila (2016) | $83.45 \pm 0.29$ | - |

Table 4: Accuracy on SVHN test set for 500 labeled examples in the training. Presented results include the average and standard deviation over 5 runs with random sampling of the labeled examples among the training set.

| $\mathbf{N_L}$ | ReLU-SVHN**500**, | Sigmoid-SVHN**500**, , |
|---|---|---|
| (0.7,0.1) | $88.69 \pm 0.4$ | $\mathbf{89.03} \pm 1.5$ |
| (0.7,0.2) | $88.26 \pm 1.23$ | $88.68 \pm 0.48$ |
| (0.7,0.3) | $86.89 \pm 2.1$ | $89.52 \pm 0.2$ |
| (0.8,0.1) | $\mathbf{89.75} \pm 1.2$ | $89.02 \pm 1.5$ |
| (0.8,0.2) | $87.75 \pm 1.4$ | $88.2 \pm 1$ |
| (0.8,0.3) | $88.25 \pm 1.12$ | $88.39 \pm 0.9$ |
| (0.85,1) | $80.42 \pm 2.4$ | $79.77 \pm 1.5$ |

| $\mathbf{N_L}$ | SVHN**500**, | SVHN**1000**, |
|---|---|---|
| Improved GAN Salimans et al. (2016) | $81.56 \pm 4.8$ | - |
| Auxiliary Deep Generative Model Maaløe et al. (2016) | - | 77.14 |
| Skip Deep Generative Model Maaløe et al. (2016) | - | $83.39 \pm 0.24$ |
| Virtual Adversarial Miyato et al. (2015) | - | 75.37 |
| DGN Kingma et al. (2014) | - | $63.98 \pm 0.1$ |
| Triple GAN Li et al. (2017) | - | $94.23 \pm 0.17$ |
| Semi-Sup Requires a Bad GAN Dai et al. (2017) | - | $95.75 \pm 0.03$ |
| ΠModelLaine & Aila (2016) | $92.95 \pm 0.3$ | $94.57 \pm 0.25$ |
| VATMiyato et al. (2015) | - | 75.37 |

We train here networks based on raw audio using CNNs as detailed in the appendix. We vary $(\alpha, \beta)$ over 10 runs to demonstrate that the non-regularized supervised model is not optimal. The maximum validation accuracies on the last 100 epochs (Fig. 2) show that the regularized networks tend to learn more slowly, but always generalize better than the not regularized baseline ($\alpha = 1, \beta = 0$).

## 4 CONCLUSIONS AND FUTURE WORK

We have presented a well-justified inversion scheme for deep neural networks with an application to semi-supervised learning. By demonstrating the ability of the method to best current state-of-the-art results on MNIST with different possible topologies support the portability of the technique as well as its potential. These results open up many questions in this yet undeveloped area of DNN inversion, input reconstruction, and their impact on learning and stability.

Among the possible extensions, one can develop the reconstruction loss into a per-layer reconstruction loss. Doing so, there is the possibility to weight each layer penalty bringing flexibility as well as meaningful reconstruction. Define the per layer loss as

$$\mathcal{L}(X_n, Y_n) = \alpha L_{CE}(Y_n, \hat{y}(X_n)) 1_{\{Y_n \neq \emptyset\}} + \beta E(X_n) 1_{\{Y_n = \emptyset\}} + \sum_{\ell=0}^{L-1} \gamma^{(\ell)} R^{(\ell)}(X_n), \qquad (11)$$

with

$$R^{(\ell)}(X_n) = ||(\frac{df_\Theta(X_n)}{d\mathbf{z}^{(\ell)}(X_n)})^T f_\Theta(X_n) - \mathbf{z}^{(\ell)}(X_n)||^2. \qquad (12)$$

Doing so, one can adopt a strategy in favor of high reconstruction objective for inner layers, close to the final latent representation $\mathbf{z}^{(L)}$ in order to lessen the reconstruction cost for layers closer to the input $X_n$. In fact, inputs of standard dataset are usually noisy, with background, and the object of interest only contains a small energy with respect to the total energy of $X_n$. Another extension would be to update the weighting while performing learning. Hence, if we denote by $t$ the position

| $\alpha$ | $\beta$ | train mean (std) | valid mean (std) | test mean (std) |
|---|---|---|---|---|
| **1** | **0** | 90.41 (4.106) | 53.01 (2.534) | **51.40** (1.198) |
| 0.7 | 0.5 | still being computed computing | still being computed | still being computed |
| 0.7 | 0.4 | 91.88 (1.041) | 51.76 (1.401) | 52.23 (1.700) |
| 0.7 | 0.3 | 92.24 (1.166) | 52.45 (2.853) | 51.53 (1.441) |
| 0.7 | 0.2 | 91.56 (1.287) | 52.24 (1.935) | 51.77 (1.826) |
| 0.7 | 0.1 | 91.15 (1.101) | 52.87 (3.538) | 52.04 (1.738) |
| 0.6 | 0.5 | still being computed | still being computed | still being computed |
| 0.6 | 0.4 | 91.63 (1.391) | 51.97 (3.316) | 52.00 (1.756) |
| 0.6 | 0.3 | 91.51 (1.259) | 51.43 (1.933) | **52.95** (1.239) |
| 0.6 | 0.2 | 92.41 (0.9555) | 50.32 (1.207) | 52.07 (1.216) |
| 0.6 | 0.1 | 90.27 (3.050) | 53.11 (3.604) | 51.74 (1.631) |
| 0.5 | 0.5 | still being computed | still being computed | still being computed |
| 0.5 | 0.4 | 91.52 (0.9859) | 52.21 (2.968) | 51.92 (0.990) |
| 0.5 | 0.3 | 90.50 (1.341) | 52.04 (3.335) | 51.18 (0.937) |
| 0.5 | 0.2 | 90.13 (1.901) | 52.71 (3.431) | 51.46 (1.390) |
| 0.5 | 0.1 | 88.52 (3.152) | 53.86 (3.579) | 51.19 (1.479) |

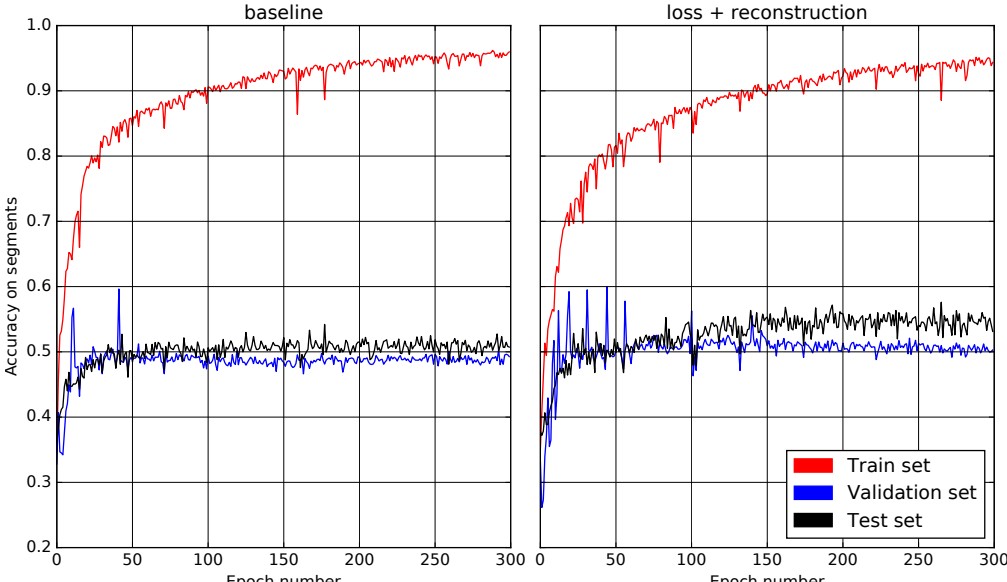

Figure 2: Top: Accuracy for Bird10 baseline ($\alpha = 1, \beta = 0$), versus regularized networks. Bottom Left: Learning curves of the baseline model. Bottom Right: Learning curve of the best test ($\alpha = 0.7$, $\beta = 0.3$).

in time such as the current epoch or batch, we now have the previous loss becoming

$$\mathcal{L}(X_n, Y_n; \Theta) = \alpha(t)L_{CE}(Y_n, \hat{y}(X_n))1_{\{Y_n \neq \emptyset\}} + \beta(t)E(X_n)1_{\{Y_n = \emptyset\}} + \sum_{\ell=0}^{L-1} \gamma^{(\ell)}(t)R^{(\ell)}(X_n). \tag{13}$$

One approach would be to impose some deterministic policy based on heuristic such as favoring reconstruction at the beginning to then switch to classification and entropy minimization. Finer approaches could rely on explicit optimization schemes for those coefficients. One way to perform this, would be to optimize the loss weighting coefficients $\alpha, \beta, \gamma^{(\ell)}$ after each batch or epoch by backpropagation on the updates weights. Define

$$\Theta(t+1) = \Theta(t) - \lambda \frac{dL(X_n, Y_n)}{d\Theta}, \tag{14}$$

as a generic iterative update based on a given policy such as gradient descent. One can thus adopt the following update strategy for the hyper-parameters as

$$\gamma^{(\ell)}(t+1) = \gamma^{(\ell)}(t) - \frac{dL(X_n, Y_n; \Theta(t+1))}{d\gamma^{(\ell)}(t)}, \tag{15}$$

and so for all hyper-parameters. Another approach would be to use adversarial training to update those hyper-parameters where both update cooperate trying to accelerate learning.

EBGAN (Zhao et al. (2016)) are GANs where the discriminant network $D$ measures the energy of a given input $X$. $D$ is formulated such as generated data produce high energy and real data produce lower energy. Same authors propose the use of an auto-encoder to compute such energy function. We plan to replace this autoencoder using our proposed method to reconstruct $X$ and compute the energy; hence $D(X) = R(X)$ and only one-half the parameters will be needed for $D$.

Finally, our approach opens the possibility of performing unsupervised tasks such as clustering. In fact, by setting $\alpha = 0$, we are in a fully unsupervised framework. Moreover, $\beta$ can push the mapping $f_\Theta$ to produce a low-entropy, clustered, representation or rather simply to produce optimal reconstruction. Even in a fully unsupervised and reconstruction case ($\alpha = 0, \beta = 1$), the proposed framework is not similar to a deep-autoencoder for two main reasons. First, there is no greedy (per layer) reconstruction loss, only the final output is considered in the reconstruction loss. Second, while in both case there is parameter sharing, in our case there is also "activation" sharing that corresponds to the states (spline) that were used in the forward pass that will also be used for the backward one. In a deep autoencoder, the backward activation states are induced by the backward projection and will most likely not be equal to the forward ones.

## 5 ACKNOWLEDGMENT

We thank PACA region and NortekMed, and GDR MADICS CNRS EADM action for their support.

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

## A   MODELS AND TRAINING DESCRIPTION

For all the experiments, the used DNN have the following number of free parameters : Resnet2-32:330K, Resnet3-16:136K, SmallCNN:131K,LargeCNN:1370K.

### A.1   MNIST

We trained the models in Tab. 2 by cross validation of $\alpha, \beta$ with step size of 0.1. The learning rate with Adam was optimized for MNIST semisup to 0.01 for SmallCNN and 0.01/3 for Resnet and LargeCNN.

### A.2   BIRD10

We trained the models in Fig. 2 as follow: conv1d (with kernel size of 1024, stride of 512 and number of filters of 42) -> maxpool1d(4, 2) -> conv1d (3, 2, 42) -> Dense(32) -> dropout(0.5) -> Dense(10). For this experiment we measured the accuracy per segment using segments of 500ms with an overlap of 90 percent.

Note that we did not used batch normalization in any of our models, because it would perturb the training as the update of the parameters will take effect before the update of normalization. We are working on a new procedure to update the parameters to overpass this issue.

## A.3  MNIST RECONSTRUCTION ON OTHER TOPOLOGIES

We give below the figures of the reconstruction of the same test sample by four different nets :
LargeUCNN ($\alpha = 0.5, \beta = 0.5$), SmallUCNN (0.6,0.5), Resnet2-32 (0.3,0.5), Resnet3-16 (0.6,0.5).
The columns from left to right correspond to: the original image, mean-pooling reconstruction, max-pooling reconstruction, and inhibitor connections. One can see that our network is able to correctly
reconstruct the test sample.

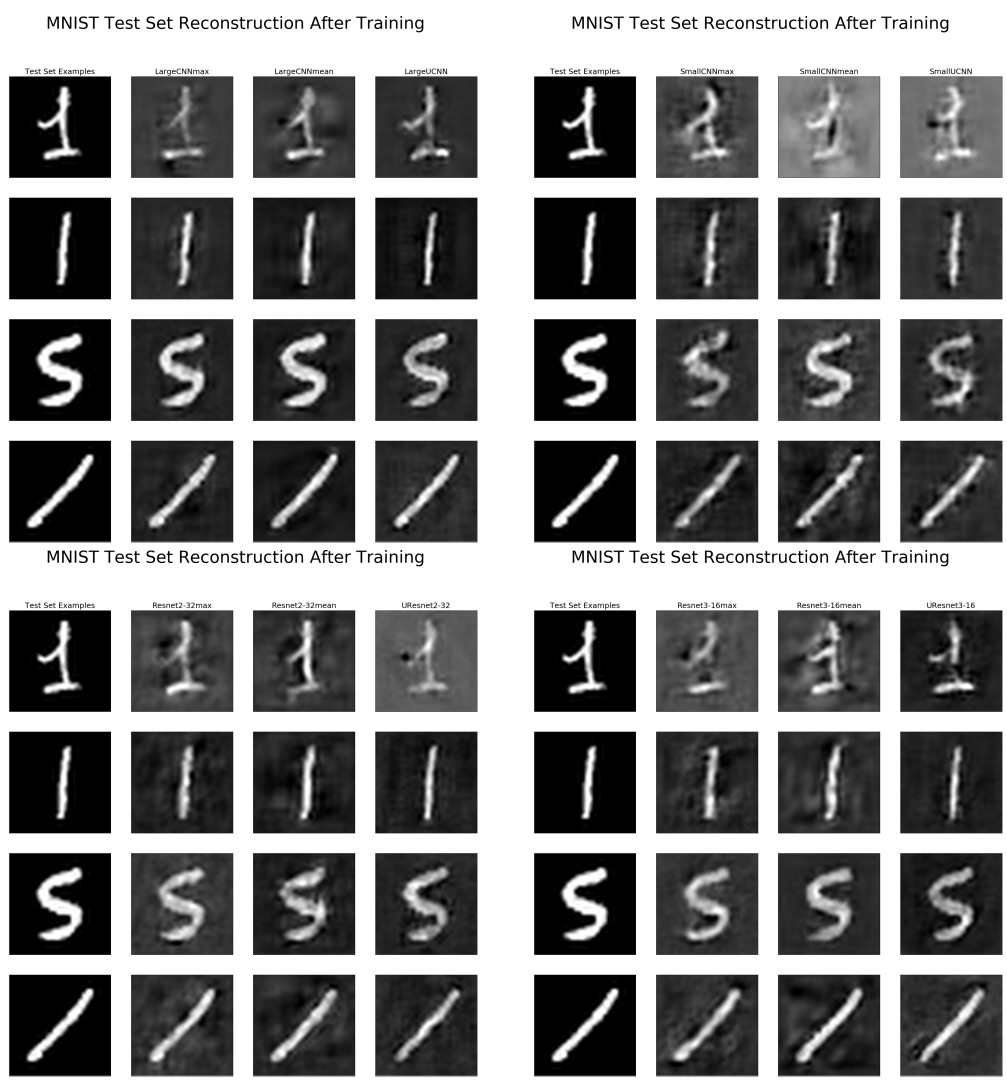

Figure 3: Reconstruction of the studied DNN models, from top to right : LargeUCNN ($\alpha = 0.5, \beta = 0.5$), SmallUCNN (0.6,0.5), Resnet2-32 (0.3,0.5), Resnet3-16 (0.6,0.5) on the same test set samples.
The columns from left to right correspond to: the original image, mean-pooling reconstruction, max-pooling reconstruction, inhibitor connections.

