# OpenReview forum: "Semi-Supervised Learning via New Deep Network Inversion"
_ICLR.cc/2018/Conference — Reject_

### Official Review · AnonReviewer3 · 2017-11-22
**plausible idea needs better motivation and demonstration**

**Rating:** 5
**Confidence:** 4

**Review:**

After reading the revision:

The authors addressed my detailed questions on experiments. It appears sometimes the entropy loss (which is not the main contribution of the paper) is essential to improve performance; this obscures the main contribution.

On the other hand, the theoretical part of the paper is not really improved in my opinion, I still can not see how previous work by Balestriero and Baraniuk 2017 motivates and backups the proposed method.

My rating of this paper would remain the same.

============================================================================================

This paper propose to use the reconstruction loss, defined in a somewhat unusual way, as a regularizar for semi-supervised learning.

Pros:

The intuition is that the ReLU network output is locally linear for each input, and one can use the conjugate mapping (which is also linear) for reconstructing the inputs, as in PCA. Realizing that the linear mapping is the derivative of network output w.r.t. the input (the Jacobian), the authors proposed to use the reconstruction loss defined in (8). Different from typical auto-encoders, this work does not require another reconstruction network, but instead uses the "derivative".  This observation is neat in my opinion, and does suggest a different use of the Jacobian in deep learning.  The related work include auto-encoders where the weights of symmetric layers are tied.

Cons:

The motivation (Section 2) needs to be improved. In particular, the introduction/review of the work of Balestriero and Baraniuk 2017 not very useful to the readers.  Notations in eqns (2) and (3) are not fully explained (e.g., boldface c). Intuition and implications of Theorem 1 is not sufficiently discussed: what do you mean by optimal DNN, what is the criteria for optimality? is there a generative assumption of the data underlying the theorem? and the assumption of all samples being norm 1 seems too strong and perhaps limits its application? As far as I see, section 2 is somewhat detached from the rest of the paper.

The main contribution of this paper is supposed to be the reconstruction mapping (6) and its effect in semi-supervised learning. The introduction of entropy regularization in sec 2.3 seems somewhat odd and obscures the contribution. It also bears the questions that how important is the entropy regularization vs. the reconstruction loss. In experiments, results with beta=1.0 need to be presented to assess the importance of network inversion and the reconstruction loss. Also, a comparison against typical auto-encoders (which uses another decoder networks, with weights possibly tied with the encoder networks) is missing.

---

> ### Author Response · Authors · 2017-12-16
> **Updated paper and answers**
>
> Thanks for your review and questions. Concerning the comparison with pure auto-encoders, we refer to the ladder network in our related work which is a special case of auto-encoder used for semi-supervised learning.
>
> We have added additional motivation for our loss functions and their implications.  In particular, we have provided two new experiments on CIFAR10 and SVHN with the hyper-parameters $\beta=1$, which eliminates the entropy term from the loss function. As you can see from the new experiments, the entropy loss contributes significantly on SVHN but not on CIFAR10.

---

### Official Review · AnonReviewer1 · 2017-11-24
**Some comments on the related work, motivation and experiments**

**Rating:** 4
**Confidence:** 5

**Review:**

In summary, the paper is based on a recent work Balestriero & Baraniuk 2017 to do semi-supervised learning. In Balestriero & Baraniuk, it is shown that any DNN can be approximated via a linear spline and hence can be inverted to produce the "reconstruction" of the input, which can be naturally used to do unsupervised or semi-supervised learning. This paper proposes to use automatic differentiation to compute the inverse function efficiently. The idea seems interesting. However, I think there are several main drawbacks, detailed as follows:

1. The paper lacks a coherent and complete review of the semi-supervised deep learning. Herewith some important missing papers, which are the previous or current state-of-the-art.

[1] Laine S, Aila T. Temporal Ensembling for Semi-Supervised Learning[J]. arXiv preprint arXiv:1610.02242, ICLR 2016.
[2] Li C, Xu K, Zhu J, et al. Triple Generative Adversarial Nets[J]. arXiv preprint arXiv:1703.02291, NIPS 2017.
[3] Dai Z, Yang Z, Yang F, et al. Good Semi-supervised Learning that Requires a Bad GAN[J]. arXiv preprint arXiv:1705.09783, NIPS 2017.

Besides, some papers should be mentioned in the related work such as Kingma et. al. 2014. I'm not an expert of the network inversion and not sure whether the related work of this part is sufficient or not.

2. The motivation is not sufficient and not well supported.

As stated in the introduction, the authors think there are several drawbacks of existing methods including "training instability, lack of topology generalization and computational complexity." Based on my knowledge, there are two main families of semi-supervised deep learning methods, classified by depending on deep generative models or not.  The generative approaches based on VAEs and GANs are time consuming, but according to my experience, the training of VAE-based methods are stable and the topology generalization ability of such methods are good. Besides, the feed-forward approaches including [1] mentioned above are efficient and not too sensitive with respect to the network architectures.  Overall, I think the drawbacks mentioned in the paper are not common in existing methods and I do not see clear benefits of the proposed method. Again, I strongly suggest the authors to provide a complete review of the literature.

Further, please explicitly support your claim via experiments. For instance, the proposed method should be compared  with the discriminative approaches including VAT and [1] in terms of the training efficiency. It's not fair to say GAN-based methods require more training time because these methods can do generation and style-class disentanglement while the proposed method cannot.

3. The experimental results are not so convincing.

First, please systematically compare your methods with existing methods on the widely adopted benchmarks including MNIST with 20, 100 labels and SVHN with 500, 1000 labels and CIFAR10 with 4000 labels. It is not safe to say the proposed method is the state-of-the-art by only showing the results in one setting.

Second, please report the results of the proposed method with comparable architectures used in previous methods and state clearly the number of parameters in each model. Resnet is powerful but previous methods did not use that.

Last, show the sensitive results of the proposed method by tuning alpha and beta. For instance, please show what is the actual contribution of the proposed reconstruction loss to the classification accuracy with the other losses existing or not?

I think the quality of the paper should be further improved by addressing these problems and currently it should be rejected.

---

> ### Author Response · Authors · 2017-12-16
> **Updated paper and Answers**
>
> Thanks for your insightful comments and suggestions. The paper has been updated with the following changes.
>
> The related work section has been completed by adding recent published papers on semi-supervised learning. After a careful review of recent GAN literature, we think that our concerns about GANs for semi-supervised learning were incorrect; these comments have thus been removed.
>
> However, we emphasize that the primary aim of the paper is a new signal reconstruction (inversion) method for a broad range of deep nets. The application to semi-supervised learning (for which we reach very reasonable results with very few changes to the supervised learning ML pipeline) is secondary.
>
> Concerning the experiments, we have updated the paper with new experiments on SVHN and CIFAR10 to support our approach (we expect the new experiments to be fully completed in 20 days). We have limited our experiments to the CNN topology with ReLU and sigmoid nonlinearities in order to provide a more clear comparison with the literature (the literature has not yet applied Resnets to semi-supervised learning). We also added details on the number of parameters of each model in the appendix.

---

> > ### Comment · AnonReviewer1 · 2018-01-01
> > **Limited contribution**
> >
> > Thanks for the rebuttal and revision. Based on the complete survey and experiments, I think the contribution of this paper is not significant. Intuitively, I cannot see why this kind of reconstruction is more useful for deep learning, compared with existing network inversion methods. Practically, semi-supervised learning is the most important scenario in this paper while the results in this setting are not competitive to the state-of-the-art in two real datasets. Therefore, I stand by my reviews. BTW, it's better to improve the layout of the article, especially the tables.

---

> > > ### Author Response · Authors · 2018-01-05
> > > **Answer Reviewer1**
> > >
> > > We thank you for your constructive comments and targeted concerns. We performed a clarification of the tables. Concerning the semi-supervised results, given the differences between our new approach (which uses a given ‘fixed’ training set and our new inversion formula that works for arbitrary deep networks) vs. GANs (which use an ‘unlimited’ training set), we feel that a difference of less than 3% on SVHN and 5% on CIFAR is reasonable. Indeed, the semi-supervised classification results are not the main goal of the paper. They are mainly to support our main goal of developing an inversion formula for arbitrary deep networks, which has many applications beyond semi-supervised learning. The paper also demonstrates how our inversion formula enables semi-supervised learning with Resnets, which is totally novel. To summarize, the goal of this paper is not to merely improve current approaches to semi-supervised learning but to open up a new way to work with deep nets of all kinds.
> > > Regards

---

### Official Review · AnonReviewer2 · 2017-11-26
**SEMI-SUPERVISED LEARNING VIA NEW DEEP NETWORK INVERSION**

**Rating:** 7
**Confidence:** 2

**Review:**

This paper proposed a new optimization framework for semi-supervised learning based on derived inversion scheme for deep neural networks. The numerical experiments show a significant improvement in accuracy of the approach.

---

> ### Author Response · Authors · 2017-12-16
> **Openings**
>
> Thanks for reviewing our work. We invite additional comments as the review period progresses; we will do our best to respond asap.

---

### Decision · Program_Chairs · 2018-01-29
**ICLR 2018 Conference Acceptance Decision**

**Decision:**

Reject

**Comment:**

The paper proposes a novel approach for DNN inversion mainly targeted towards semi-supervised learning. However the semi-supervised learning results are not competitive enough. Although the authors mention in the author-response that semi-supervised learning is not the main goal of the paper, the experiments and claims of the paper are mainly targeted towards semi-supervised learning. As the approach for inversion is novel, the paper could be motivated from a different angle with appropriate supporting experiments. In its current form it's not suitable for publication.